# Entropy Scheduling in Reinforcement Learning for Large Language Models

## Abstract

We observe that *entropy* in reinforcement learning functions analogously to the *learning rate* in LLMs. Maintaining stable entropy, as demonstrated in DAPO (Yu et al., 2025), helps stabilize RL training, while rapid entropy annealing (i.e., so-called entropy collapse) accelerates local performance improvement and enables faster convergence. We argue that these two processes are not antithetical, but can be effectively controlled and scheduled within a single training run, similar to learning rate scheduling. We propose *Entropy Schduling* (ES), which optimizes different pre-set goals (e.g. $k$ in optimizing Pass@$k$) by controlling and scheduling entropy at each step of the RL process. We find that maintaining stable entropy early in training followed by entropy annealing achieves superior performance. Moreover, since stable-state entropy and annealed entropy exhibit distinctly different learning dynamics, curriculum learning can be seamlessly integrated to maximize model performance based on different entropy phases. We show that entropy scheduling is straightforward to implement and intuitive in design. Extensive experiments suggest that it delivers consistent and stable performance improvements across diverse models and algorithms.

## 1 Introduction

The field of reinforcement learning with verifiable rewards (RLVR) for large language models (LLMs) has witnessed rapid advancements, as illustrated by strong reasoning models such as DeepSeek-R1(DeepSeek-AI et al., 2025), Kimi-K1.5(Team et al., 2025), and the OpenAI o-series (OpenAI, 2024; 2025), which demonstrate significant improvements in reasoning capabilities. A critical aspect of RLVR is the *policy entropy*, which quantifies the unpredictability of the policy distribution. High entropy denotes that the policy model samples actions more uniformly with greater randomness, while low entropy suggests the policy is concentrating on a narrower set of actions. Entropy collapse (or entropy annealing), a phenomenon commonly observed during RL for LLMs, occurs when the policy entropy diminishes too rapidly, limiting exploration. For example, vanilla GRPO (Shao et al., 2024), a widely used RL algorithm for large reasoning models, has been shown to cause entropy collapse during

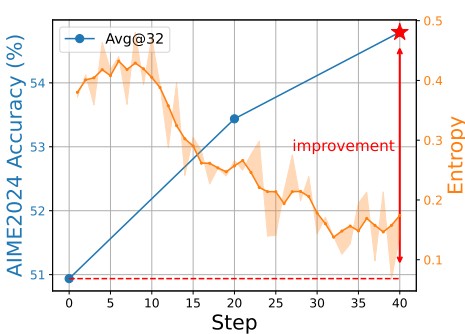

Figure 1: We train the DAPO-32B (Yu et al., 2025) with a lower clip value and the same dataset to achieve entropy annealing, and improve AIME2024 from 50.9 to 54.9 within 40 training steps.

RLVR training (Yu et al., 2025). To mitigate this, several strategies have been proposed (Yu et al., 2025; Cui et al., 2025) to maintain entropy at relatively high levels (referred to as stable entropy). Elevated and stable entropy levels are known to encourage extensive exploration, enabling the model to generate diverse reasoning outputs and stabilizing the training process.

In contrast to traditional perspectives, very recent studies demonstrate that unsupervised learning in RLVR, driven solely by entropy minimization, can significantly enhance model performance within a short, localized training step (Wang et al., 2025c; Agarwal et al., 2025; Gao et al., 2025). As a pilot

study, we continually train and anneal the entropy of `DAPO-Qwen-32B` [1], which is a well trained model ending with high entropy (Yu et al., 2025). As illustrated in Fig. 1, this approach achieved a 4.0% absolute performance improvement in merely 40 training steps.

We reconcile these two seemingly opposing perspectives and optimization methods. We find that entropy resembles the learning rate in LLM training. Maintaining stable entropy contributes to the long-term sustained development of RL training, while subsequent entropy decay or annealing helps the model rapidly improve performance and converge in the short term. As a summary, entropy that stabilizes first and then decays, compared to entropy that decays first and then stabilizes, exhibits weaker performance initially but stronger performance in the end. We refer to this empirical law as *the Parallelogram Law of Entropy*.

On the other hand, we also find that what makes entropy more compelling than the learning rate is that many important hyperparameters in RL (e.g., clip-higher coefficient (Yu et al., 2025), entropy loss coefficient, and KL penalty coefficient) ultimately control the model's performance by managing entropy. In other words, entropy is the (almost only) fundamental factor in altering model performance (like exploration and explanation).

Inspired by learning rate schedule (LRS) techniques, we propose *Entropy Scheduling* (ES) to combine entropy stable stage and entropy annealing stage. Specifically, as a demonstration, we control the entropy with constant, cosine and the stable-annealing schedules. Moreover, we train the model with different stable constant values and different annealing ratios in stable-annealing schedule. We find that there exists an optimal entropy settings (e.g., constant entropy value, annealing ratio, etc.) for different pre-set goals (e.g. $k$ in optimizing Pass@$k$).

A major challenge in implementing ES is controlling entropy within expected and predefined ranges at each training step. Essentially, entropy is merely an indicator of the model's computation on the current batch of data and cannot be directly regulated. It can only be controlled indirectly by adjusting hyper-parameters (e.g., clip-higher, entropy loss coefficient) several steps to hundreds of steps in advance to keep entropy within a preset range at each training step. As such, we proposed a simple PID (Proportional-Integral-Derivative)-based delay control algorithm (Ang et al., 2005) as a baseline, which effectively achieves our entropy scheduling objectives.

Furthermore, due to the different learning dynamics between the entropy stable and entropy annealing, curriculum learning can be seamlessly integrated to maximize model performance during the entropy annealing phase. We can integrate all factors that contribute to performance improvement into the annealing phase. Specifically, high-quality data can be concentrated in the entropy annealing phase to amplify its advantages. Additionally, increasing the rollout number and extending the maximum response length during the annealing phase proves to be an effective method for boosting model performance.

## 2 PRELIMINARY

The LLMs act as the policy model $\pi_\theta$ and autoregressively generate output sequence $y = \{y_1, \cdots, y_t, \cdots, y_T\}$ given the prompt $x$. Proximal Policy Optimization (PPO) (Schulman et al., 2017) is a widely utilized algorithm in RL, known for its stability and efficiency. It employs a clipped surrogate objective function to constrain policy updates within a proximal region relative to the previous policy.

$$\mathcal{J}(\theta) = \mathbb{E} \left[ \min \left( \frac{\pi_\theta \left( y_t \mid y_{<t} \right)}{\pi_{\theta_{\text{old}}} \left( y_t \mid y_{<t} \right)} \hat{A}_t, \text{clip} \left( \frac{\pi_\theta \left( y_t \mid y_{<t} \right)}{\pi_{\theta_{\text{old}}} \left( y_t \mid y_{<t} \right)}, 1 - \varepsilon, 1 + \varepsilon \right) \hat{A}_t \right) \right] \quad (1)$$

where $\hat{A}_t$ is the advantage at time step $t$. In this work, we primarily focus on GRPO (Shao et al., 2024), a variation of PPO. GRPO estimates the advantage in a group-relative manner. For each prompt, the policy model samples a group of $n$ responses and estimate the advantage as follows:

$$\hat{A}_{i,t} = \frac{r_i - \text{mean} \left( \{R_i\}_{i=1}^n \right)}{\text{std} \left( \{R_i\}_{i=1}^n \right)} \quad (2)$$

where $\{R_i\}_{i=1}^n$ is the reward of each generated response.

---

[1]`https://huggingface.co/BytedTsinghua-SIA/DAPO-Qwen-32B`

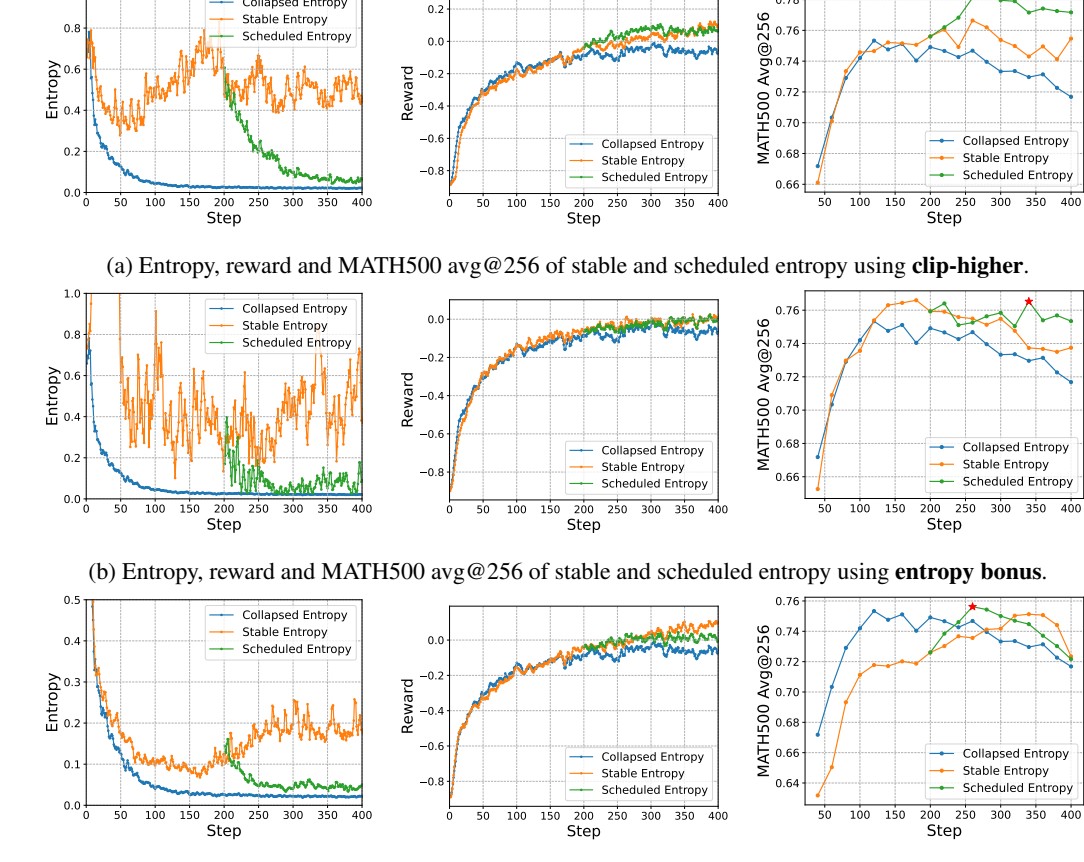

(a) Entropy, reward and MATH500 avg@256 of stable and scheduled entropy using **clip-higher**.

(b) Entropy, reward and MATH500 avg@256 of stable and scheduled entropy using **entropy bonus**.

(c) Entropy, reward and MATH500 avg@256 of stable and scheduled entropy using **KL penalty**.

Figure 2: We leverage the clip-higher, entropy bonus and KL penalty method to control different stable and scheduled entropy. All these methods or hyperparameters enable control model's reward and performance by managing entropy. The scheduled entropy that stabilizes first and then decays (orange lines) exhibits higher reward and benchmark performance compared with other scheduled entropy, which is referred as the parallelogram law of entropy.

KL penalty is used to penalize the divergence between the online policy and the reference policy (Abdolmaleki et al., 2018; Kappen et al., 2012). The objective function with KL is:

$$\mathcal{J}(\theta) = \mathbb{E}\left[\min\left(\frac{\pi_\theta\left(y_t \mid y_{<t}\right)}{\pi_{\theta_{\text{old}}}\left(y_t \mid y_{<t}\right)}\hat{A}_t, \text{clip}\left(\frac{\pi_\theta\left(y_t \mid y_{<t}\right)}{\pi_{\theta_{\text{old}}}\left(y_t \mid y_{<t}\right)}, 1-\varepsilon, 1+\varepsilon\right)\hat{A}_t\right) - \beta D_{\text{KL}}\left(\pi_\theta \| \pi_{\text{ref}}\right)\right] \tag{3}$$

The entropy bonus term is used to encourage stochasticity in the optimal policy model and could be incorporated into the objective function:

$$\mathcal{J}(\theta) = \mathbb{E}\left[\min\left(\frac{\pi_\theta\left(y_t \mid y_{<t}\right)}{\pi_{\theta_{\text{old}}}\left(y_t \mid y_{<t}\right)}\hat{A}_t, \text{clip}\left(\frac{\pi_\theta\left(y_t \mid y_{<t}\right)}{\pi_{\theta_{\text{old}}}\left(y_t \mid y_{<t}\right)}, 1-\varepsilon, 1+\varepsilon\right)\hat{A}_t\right) + \alpha\mathcal{H}\left(\pi_\theta, \mathcal{D}\right)\right] \tag{4}$$

where

$$\mathcal{H}\left(\pi_\theta, \mathcal{D}\right) = -\mathbb{E}_{\mathcal{D}, \pi_\theta}\left[\log \pi_\theta\left(y_t \mid \boldsymbol{y}_{<t}\right)\right] \tag{5}$$

Several factors influence the entropy dynamics of policy models during training. DAPO (Yu et al., 2025) claims that clip-higher is an effective method to enhance the policy model entropy and generate more diverse samples. Similarly, entropy bonus directly encourages the policy model to maintain a high entropy value, making it one of the most straightforward methods to promote exploration. Meanwhile, the KL penalty is also an effective method to avoid the entropy collapse which is equiv-

alent to the entropy bonus to some extent (Jaques et al., 2019).

$$\mathcal{J}(\theta) = \mathbb{E}\left[ \min\left( \frac{\pi_\theta\left(y_t \mid y_{<t}\right)}{\pi_{\theta_{\text{old}}}\left(y_t \mid y_{<t}\right)} \hat{A}_t, \text{clip}\left( \frac{\pi_\theta\left(y_t \mid y_{<t}\right)}{\pi_{\theta_{\text{old}}}\left(y_t \mid y_{<t}\right)}, 1 - \varepsilon, 1 + \varepsilon \right) \hat{A}_t \right) \right.$$
$$\left. + \alpha \mathcal{H}\left(\pi_\theta, \mathcal{D}\right) - \beta D_{\text{KL}}\left(\pi_\theta \| \pi_{\text{ref}}\right) \right], \tag{6}$$

## 3 PILOT STUDY: THE PARALLELOGRAM LAW OF ENTROPY

### 3.1 EXPERIMENT SETUP

We train the models with three kinds of scheduled entropy: (1) entropy annealing from the beginning of training, (2) maintaining entropy stable throughout the entire training process, and (3) introducing entropy annealing only during the final phase of training. For entropy annealing from scratch, we adopt the GRPO method without applying any KL penalty. To achieve entropy stable, we separately employ techniques such as clip-higher, entropy bonus, and KL penalty to maintain the entropy at a relatively high and stable value. Following the stable phase, we revert to the lower clip value or remove the entropy bonus and KL penalty, allowing entropy to anneal naturally from this stable state. We utilize the Qwen-2.5-7B-Base(Qwen et al., 2025) as the initial model and train it on the DAPO-17k(Yu et al., 2025) dataset. For the clip-higher method, we set the clip value to 0.28, consistent with the configuration reported in DAPO. Additionally, we use coefficients of 0.0015 and 0.005 for entropy bonus and KL penalty, respectively. We test the model performance in the MATH500 (Lightman et al., 2023), AIME2024 and AIME2025.

### 3.2 PARALLELOGRAM LAW OF ENTROPY

DAPO (Yu et al., 2025) shows that entropy collapse induces performance degrade in the end of training. However, as shown in their paper and our experiments, the entropy collapse also leads to a better performance in the initial training steps, though much worse in the end. As illustrated in Fig. 2, entropy annealing from the beginning of training (blue line) achieves higher rewards and better performance than maintaining entropy stable (orange line) during the initial training steps. It seems to show that, lower entropy exchanges for better performance within limited training steps.

However, as entropy drops to a very low level, the model's performance eventually saturates, with only few improvements observed in later training stages. In contrast, maintaining entropy stable prevents saturation, allowing for steady improvements over time. Consequently, entropy stabilization gradually surpasses entropy annealing from scratch at longer training steps.

Now, it comes with a natural question, if we put the entropy reduction into the last stage of training, would it be beneficial to the final performance? We actually make this experiment that introducing further entropy annealing (green line) from an entropy stable phase in the final training stage. As shown in Fig. 2, this entropy annealing yields rapid improvement in both rewards and accuracy, outperforming models trained with stable entropy in same training steps. This finding aligns with concepts in learning rate annealing, where results in the rapid validation loss drop in the final learning rate annealing training stage.

Therefore, synthesizing these phenomena, here comes a conclusion: the same entropy reduction, is far more effective near the end of training than in the early stages. Figuratively speaking, the entropy annealing from the beginning line (blue line), stable entropy line (orange line) and entropy annealing in the final phase line (green line) form a parallelogram. The performance at the top line of parallelogram is initially worse but ultimately better than the performance of the bottom line of parallelogram. It is worth noting that, we not only adopt the clip-higher strategy to control entropy but also use a more direct entropy bonus and KL penalty term to control entropy, achieving quite similar effects.

### 3.3 WHAT HAPPENS IN ENTROPY ANNEALING

The trade-off between exploration and exploitation is a fundamental concept in reinforcement learning, where entropy serves as a key mechanism to regulate the transition between these two stages.

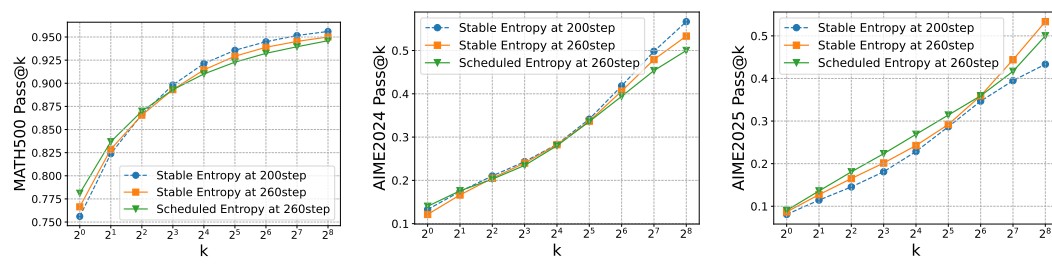

(a) Pass@$k$ curve of MATH500.  (b) Pass@$k$ curve of AIME2024.  (c) Pass@$k$ curve of AIME2025.

Figure 3: The Pass@$k$ curves reveal a distinct trade-off between annealing entropy and stable entropy. Specifically, by comparing the stable entropy (orange line) and annealing entropy (green line) with same training steps, we find that entropy annealing sacrifices the Pass@$k$ values for larger $k$ (exploration) to improve the Pass@$k$ values for smaller $k$ (exploitation).

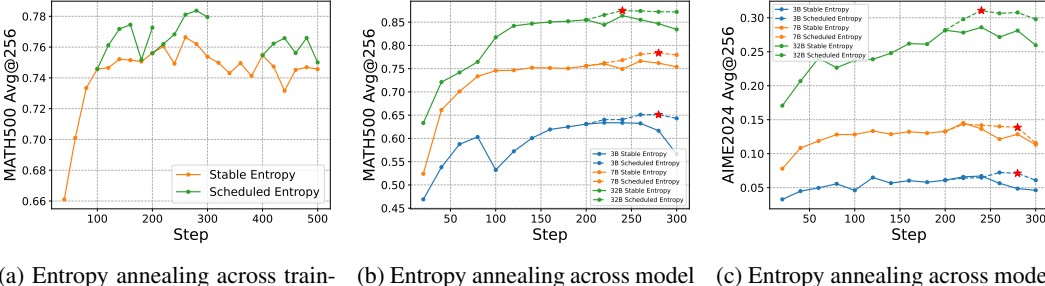

(a) Entropy annealing across training steps in MATH500.  (b) Entropy annealing across model sizes in MATH500.  (c) Entropy annealing across model sizes in AIME2024.

Figure 4: The benchmark metric of scheduled entropy across longer training steps and larger model sizes. We show that all training steps and model sizes could benefit from entropy annealing.

We evaluate the Pass@$k$ curve to quantitatively describe the exploration and exploitation of the policy models. Firstly, as depicted in Fig. 3, the RLVR process will gradually reduce the model's Pass@$k$ score of large $k$ (referred to exploration ability) but increase the model's Pass@$k$ score of small $k$ (referred to exploitation ability) by comparing the stable entropy at different training steps (blue and orange lines). Secondly, we compare the Pass@$k$ curves for entropy stable and entropy annealing under identical training steps and datasets. The entropy annealing achieves higher Pass@$k$ values compared to entropy stable for smaller values of $k$, but stable entropy exhibits higher Pass@$k$ values for larger values of $k$. This phenomenon suggests that entropy annealing sacrifices performance on larger $k$ (exploration) to enhance performance on smaller $k$ (exploitation), which explains that the annealing entropy phase achieve a higher reward or Pass@1 score.

### 3.4 ENTROPY ANNEALING FROM DAPO-32B

DAPO-32B (Yu et al., 2025) model is trained using the clip-higher method, maintaining a high entropy value throughout the entire training process. To further investigate the impact of entropy annealing, we introduce an entropy annealing phase for DAPO-32B by reducing the clipping value from 0.28 to 0.2. As depicted in Fig. 1, applying entropy annealing to DAPO-32B results in a rapid performance improvement on the AIME2024 benchmark, with performance increasing from 50.9 to 54.9 within just 40 training steps using the same training dataset. This result also highlights the effectiveness of putting the entropy reduction or annealing into the last stage of training to achieve a rapid performance improvement over relatively short training durations.

### 3.5 SCALING ON TRAINING STEPS AND MODEL SIZES

We validate the effectiveness of entropy annealing across extended training steps and varying model sizes. Specifically, we apply entropy annealing over longer training durations and observe consistent performance improvements as shown in Fig. 4a. Furthermore, we evaluate the impact of entropy

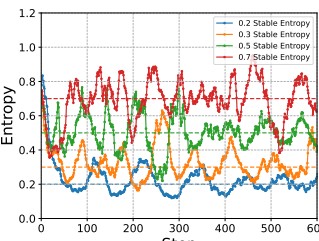 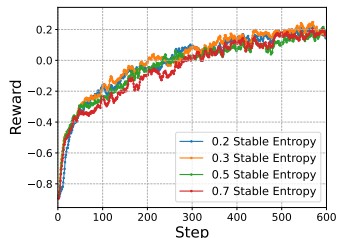 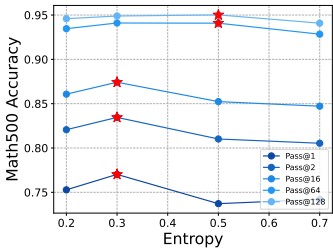

(a) Entropy of constant ES with different values.

(b) Reward of constant ES with different values.

(c) MATH500 of constant ES with different values.

Figure 5: The entropy, reward and performance of different models trained with different constant entropy values. There exist an optimal constant entropy value for different Pass@$k$ objective.

annealing on models of different sizes (3B, 7B, and 32B), maintaining consistent training steps and using the same training dataset across experiments. As illustrated in Fig. 4b and Fig. 4c, all model sizes benefit from entropy annealing, achieving performance improvement. Even larger models, which typically exhibit higher baseline performance, demonstrate a certain degree of improvement, highlighting the generalization of entropy annealing.

## 4 ENTROPY SCHEDULING

Similar to learning rate scheduling, balancing between the entropy stable phase and the annealing phase is essential for achieving optimal performance (Tissue et al., 2024; Wang et al., 2025b). Entropy stable facilitates steady reward improvements, analogous to the behavior of a constant learning rate, while entropy annealing enables rapid performance improvement, akin to learning rate annealing. Based on this analogy, we propose *Entropy Scheduling (ES)* to strategically plan the entropy dynamics in the training. However, there is actually a significant difference between learning rate and entropy. Essentially, different with learning rate as a hyper-parameter, entropy is merely an indicator of the model's computation on the current batch of data and cannot be directly regulated. It can only be controlled indirectly by adjusting hyper-parameters (e.g., clip-higher, entropy loss coefficient), and this impact on entropy does not really take effect immediately until the actor model is considerably changed after some training steps. Inspired by the PID (Proportional-Integral-Derivative) algorithm (Ang et al., 2005) in the field of automatic control, we propose an adaptive clip-higher method as a simple PID algorithm to control the entropy at preset scheduling values. We further conduct experiments using various ES methods, including constant ES, cosine ES, stable with annealing ES and cyclic ES which are all common in the LRS.

### 4.1 ADAPTIVE CLIP-HIGHER

As shown in Fig. 2, although these there methods (clip-higher, entropy bonus and KL penalty) could effectively control the policy entropy, the entropy bonus and KL penalty are extremely sensitive to the value of the coefficients. Clip-higher is a more proper method to control the policy entropy. However, existing implementations of clip typically apply a fixed clip value throughout the entire training process. During the stable entropy phase, a fixed higher clip value may fail to consistently maintain entropy at a stable level and cannot precisely determine the specific constant entropy value. Similarly, during the entropy annealing phase, using a fixed lower clip will lead to entropy decrease at a fixed rate, limiting flexibility and adaptability.

We propose the *adaptive clip-higher* approach, which is adjusted real-time in order to control entropy as scheduled value at each step. The details of the *adaptive clip-higher* are provided in Algorithm 1. Our method is a simple PID controller that adjusts the control variable (clip value) by computing the target entropy error. Firstly, the policy entropy for each training step is pre-specified according to the different ES strategy similar with LRS. Then, a sliding window is employed to compute the average actual entropy over recent training steps, which is then compared with the averaged pre-specified entropy. Based on the deviation between the actual and pre-specified entropy values, the clip value is adjusted dynamically by adding or subtracting a fixed clip offset.

---

**Algorithm 1** Adaptive Clip-Higher Algorithm

---

**Require:** Warm-up steps $T \in \mathbb{N}$, sliding window size $w \in \mathbb{N}$, adjustment step $\delta \in \mathbb{R}^+$, initial clip value $c_0 \in \mathbb{R}^+$, feasible range $[c_{\min}, c_{\max}] \subset \mathbb{R}^+$
 1: Initialize $c_t \leftarrow c_0, t \leftarrow 0$
 2: **while** training continues **do**
 3:     $t \leftarrow t + 1$
 4:     **if** $t \leq T$ **then**
 5:         $c_t \leftarrow c_0$                                                                                                     ▷ Warm-up phase
 6:     **else**
 7:         $\mathcal{W}_t \leftarrow \{t - w + 1, t - w + 2, \ldots, t\}$                                       ▷ Sliding window
 8:         $H_{\text{sched}}(t) \leftarrow \text{EntropyScheduler}(\mathcal{W}_t)$
 9:         $H_{\text{curr}}(t) \leftarrow \frac{1}{w} \sum_{\tau \in \mathcal{W}_t} H(\tau)$
10:         $\Delta c \leftarrow \begin{cases} +\delta & \text{if } H_{\text{sched}}(t) > H_{\text{curr}}(t) \\ -\delta & \text{if } H_{\text{sched}}(t) < H_{\text{curr}}(t) \\ 0 & \text{otherwise} \end{cases}$
11:         $c_t \leftarrow \text{clamp}(c_t + \Delta c, c_{\min}, c_{\max})$
12:     **end if**
13:     Execute training step with clip parameter $c_t$
14: **end while**

---

## 4.2 DIFFERENT ENTROPY SCHEDULING

After we have figured out how to control entropy, the next step is find the most appropriate scheduling. We conduct some ES experiments, like constant, cosine, and first stable then annealing schedules which are all most popular LRS in LLM training.

**Constant Entropy Scheduling**   We train the models using constant ES with varying constant entropy values. As shown in Fig. 5a, the adaptive clip-higher method efficiently controls entropy to oscillate around the pre-specified constant values. We observe that different constant entropy values lead to varying levels of performance under the same training steps. Moreover, there exists an optimal constant entropy value for for different pre-set goals (e.g. $k$ in optimizing Pass@$k$) as shown in Fig. 5c. For the optimization objective of Pass@$k$ with a smaller $k$, the optimal constant value is small. For the Pass@$k$ with larger $k$, the optimal constant value gradually increases. Higher constant entropy value encourage the LLMs to generate more diverse actions. This may lead to a lower reward or Pass@1 score, but increases the probability of generating correct answers when more samples are generated. This phenomenon is analogous to the concept of the optimal maximum learning rate encountered in the pre-training of LLMs (Bjorck et al., 2024).

**Cosine Entropy Scheduling**   Cosine LRS (Loshchilov & Hutter, 2016) is one of the most widely used strategies in the pre-training of LLMs. This scheduling approach effectively balances the stable and annealing phases, thereby enhancing overall model performance. Inspired by this, we extend the cosine scheduling concept to ES and evaluate its effectiveness, as shown in Fig. 6. Our results show that the reward and MATH500 avg@256 (both are Pass@1 metric) of cosine ES is initially lower than that of constant ES during the early stages of training. However, in the later training stages, cosine ES surpasses constant ES. This behavior can be attributed to the entropy annealing component inherent in the cosine scheduling approach. Notably, this behavior is similar to relationship between cosine and constant LRS in terms of their impact on validation loss.

**Stable and Annealing Scheduling**   WSD (Hu et al., 2024) is an effective LRS strategy that stabilizes training by maintaining a constant learning rate and then accelerates convergence through a final learning rate annealing phase. We control the entropy similar with WSD, that the entropy remain a stable during most of the training phase and then annealing in the final stage.

As shown in Fig. 6d, our method effectively maintains a high constant entropy during the stable phase and subsequently anneals with varying annealing steps or ratios. By comparing the final performance across different annealing ratios, we observe that there also exists an optimal annealing ratio for different Pass@$k$ optimization objective. As shown in Fig. 6f, the Pass@$k$ for large $k$ tends

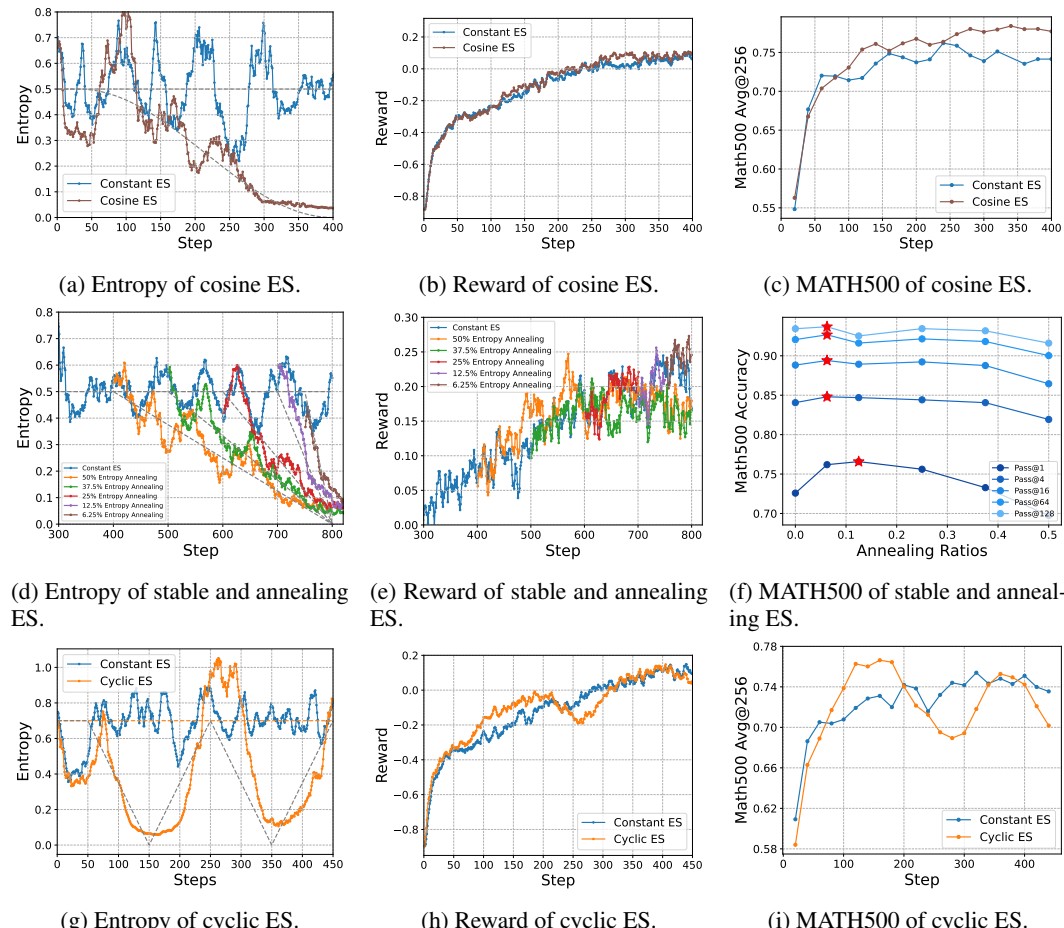

Figure 6: The model entropy, reward and MATH500 avg@256 using cosine ES, stable with annealing ES and cyclic ES. Different ES have different training dynamics in reward and performance.

to adapt a smaller annealing ratio. The smaller annealing ratio means maintaining the high entropy for a longer training process to make the model in a state of generating diversity responses, and fewer training steps to trade high Pass@$k$ performance for low Pass@$k$ performance.

**Cyclic Scheduling**  We also train the model using cyclic ES, which involves an initial entropy annealing phase followed by an increase phase. This cyclic approach aims to capture the training dynamics by alternating between low and high entropy levels.

As illustrated in Fig. 6h and Fig. 6i, during the entropy increase phase, model reward or performance improvement slows down and may even experience a decline. This behavior occurs because restoring a high-entropy state increases the model's generation diversity at the expense of exploitation ability. This is consistent with the scenario where an increasing learning rate leads to a rise in validation loss. Finally, the subsequent decrease in entropy resumes model exploitation ability and gets a nearly same performance with constant ES in the end.

## 5 ENTROPY STAGED LEARNING

The entropy stable and entropy annealing exhibit the different learning dynamics. The entropy annealing phase could solidify the model's potential for diverse reasoning generation into more stable correct reasoning generation, marking a phase of rapid performance improvement. We could maximize the advantage of entropy annealing through integrating curriculum learning, like introducing more high-quality training data and other useful RL training settings in the annealing phase.

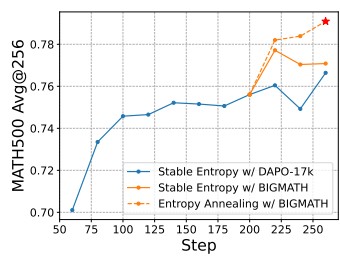 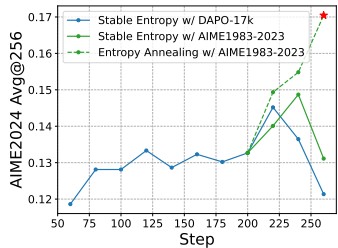 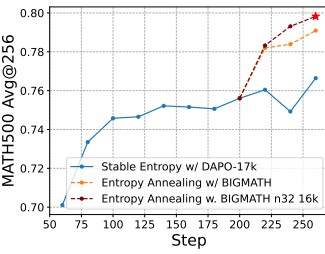

(a) MATH500 w. / wo. entropy annealing using high-quality data.

(b) AIME2024 w. / wo. entropy annealing using high-quality data.

(c) MATH500 w. lager rollout samples and context length.

Figure 7: The model performance of introducing the high-quality data and other useful RL training settings in entropy annealing phase. The entropy annealing could further maximize the advantages of high-quality data, large rollout numbers and long context length.

### 5.1 HIGH-QUALITY TRAINING DATA

In pre-training and continual pre-training of LLMs, some methods attempt to allocate high-quality data to the learning rate annealing phase to achieve better performance (Grattafiori et al., 2024; Parmar et al., 2024). This strategy is particularly effective because the quantity of high-quality data is often limited and cannot fully support the entire training process. Concentrating such data in the annealing phase can maximize its impact on model performance.

Similarly, we explore leveraging high-quality data more effectively during the entropy annealing phase to amplify its advantages. Specifically, we use AIEM1983-2023 and BigMATH (Albalak et al., 2025) datasets with difficulty levels 0-2 as the high-quality training data. As shown in Fig. 7, we compare the performance of training with high-quality data both with and without entropy annealing. Our results demonstrate that high-quality data alone (without entropy annealing) could lead to better performance compared to the baseline dataset DAPO-17k. Furthermore, incorporating entropy annealing amplifies the effect of the high-quality data, resulting in even higher accuracy.

### 5.2 INCREASING THE ROLLOUT NUMBER AND CONTEXT LENGTH

The sample number of per prompt in GRPO and maximum response length are also the important factors in RLVR training to affect the model performance. However, under limited computational resources, consistently maintaining a large rollout samples and long response length is challenging. In the continual pre-training in LLMs, the model will enlarge context length after adequate pre-training with short context. Meanwhile, gradually increasing response length is also a common method in the RL training (Luo et al., 2025).

Inspired by the better learning dynamics of entropy annealing, we could only adapt larger rollout samples and long response length in the short annealing stage. We enlarge the rollout samples from 8 to 32 and response length from 4k to 16k. As shown in Fig. 7c, we compare the original and larger rollout samples and response length in the entropy annealing stage and observe the further performance improvement.

## 6 CONCLUSION

In this work, we discovered the parallelogram law of entropy, revealing that entropy reduction is more effective at the end of RL training than in early stages, leading us to propose entropy scheduling (ES) as a simple yet powerful technique analogous to learning rate scheduling. By drawing parallels between entropy and learning rate—where high entropy/learning rate encourages global exploration while low entropy/learning rate promotes local convergence—we developed adaptive control mechanisms to maintain scheduled entropy throughout training. Our experiments demonstrate that strategic entropy scheduling, particularly with late-stage entropy reduction, can significantly improve model performance (e.g., from 50.9% to 54.9% on AIME2024 in just 40 steps), and different entropy schedules can be tailored for specific optimization goals like Pass@$k$, establishing entropy scheduling as an intuitive and effective approach for enhancing RL training in large language models.

REPRODUCIBILITY STATEMENT

Our experiments are conducted using publicly available datasets and open-source models. We provide detailed experimental setups, including hyperparameter configurations, training framework, and implementation details, in Appendix A.

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

## A  EXPERIMENTAL SETUP

We use the Qwen-2.5-Base (Qwen et al., 2025) as the initial model and use DAPO-17k (Yu et al., 2025) as training data. The training batch size is 512 and we samples 8 responses per prompt with a temperature 1. The mini batch size is 32 and we performance 16 policy updates in each training batch size. The max response length in the training is 4096. We use verl RL framework (Sheng et al., 2024) for all the training. We use the rule-based reward same with the DAPO (Yu et al., 2025):

$$R(\hat{y}, y) = \begin{cases} 1, & \text{equivalent}(\hat{y}, y) \\ -1, & \text{otherwise} \end{cases} \tag{7}$$

where $y$ is the ground-truth answer and $\hat{y}$ is the predicted answer.

In the evaluation, we also use temperature of 1 to evaluate the test accuracy. Specifically, for each problem $x_i$, we generate $n = 256$ samples and count the number of correct samples as $c_i$. We compute low-variance and unbiased Pass@$k$ estimation:

$$\text{Pass}@k := \mathbb{E}_{x_i \sim \mathcal{D}} \left[ 1 - \frac{\binom{n-c_i}{k}}{\binom{n}{k}} \right] \tag{8}$$

## B  RELATED WORK

**Reinforcement Learning for LLM**  Reinforcement learning with verifiable rewards (RLVR) has emerged as a powerful paradigm for enhancing the reasoning capabilities of large language models (Chen et al., 2025; Hu et al., 2025; He et al., 2025). DeepSeek directly uses GRPO with verifiable rewards to obtain the DeepSeek-R1 (DeepSeek-AI et al., 2025). Some recent works point out the shortcomings in the PPO(Schulman et al., 2017) or GRPO (Shao et al., 2024) algorithms, like training instability (Yu et al., 2025), reward noise (Liu et al.) and model collapse in Mixture-of-Experts (MoE) RL training (Zheng et al., 2025). We mainly focus on the entropy collapse in the RLVR training and propose the entropy scheduling to combine the stable entropy and annealing entropy to achieve better performance.

**Entropy in Reinforcement Learning**  Policy entropy is an important metric in the RLVR which could quantify the unpredictability of the policy distribution. DAPO (Yu et al., 2025) points out that the entropy collapse happened in the GRPO (Shao et al., 2024) and proposes to use a higher clip value to avoid the collapse. Some recent works (Cui et al., 2025; Wang et al., 2025a) also explain the mechanism behind entropy dynamics and propose some token-level methods to avoid the entropy collapse. Cheng et al. (2025) proposes the correlations between entropy and exploration and introduces a novel entropy related advantage function. We also uncover that entropy could control the exploration and explanation of policy models through measuring Pass@$k$ metric. What's more, we point out that the best entropy dynamics should combine stable and annealing entropy and propose the entropy scheduling which is similar with learning rate scheduling in the RL training process.

## C  RESPONSE LENGTH

We also compare the response length of stable entropy and annealing entropy phase. As shown in Fig. 8, the response length will also increase in the entropy annealing phase, which explains the performance improvement to a certain extent. What's more, the different constant ES corresponds to different response length.

## D  ADDITIONAL EXPERIMENTS

We also use other training datasets and other training features to validate the generality of ES. We use the `OpenR1-Math` [2] as the training dataset and leverage all other training settings or features

---

[2] `https://huggingface.co/datasets/open-r1/OpenR1-Math-220k`

in the DAPO (Yu et al., 2025) including token-level loss, dynamic samples and overlong reward to train the model. As shown in Fig. 9, the ES could effectively apply to other training dataset and training features.

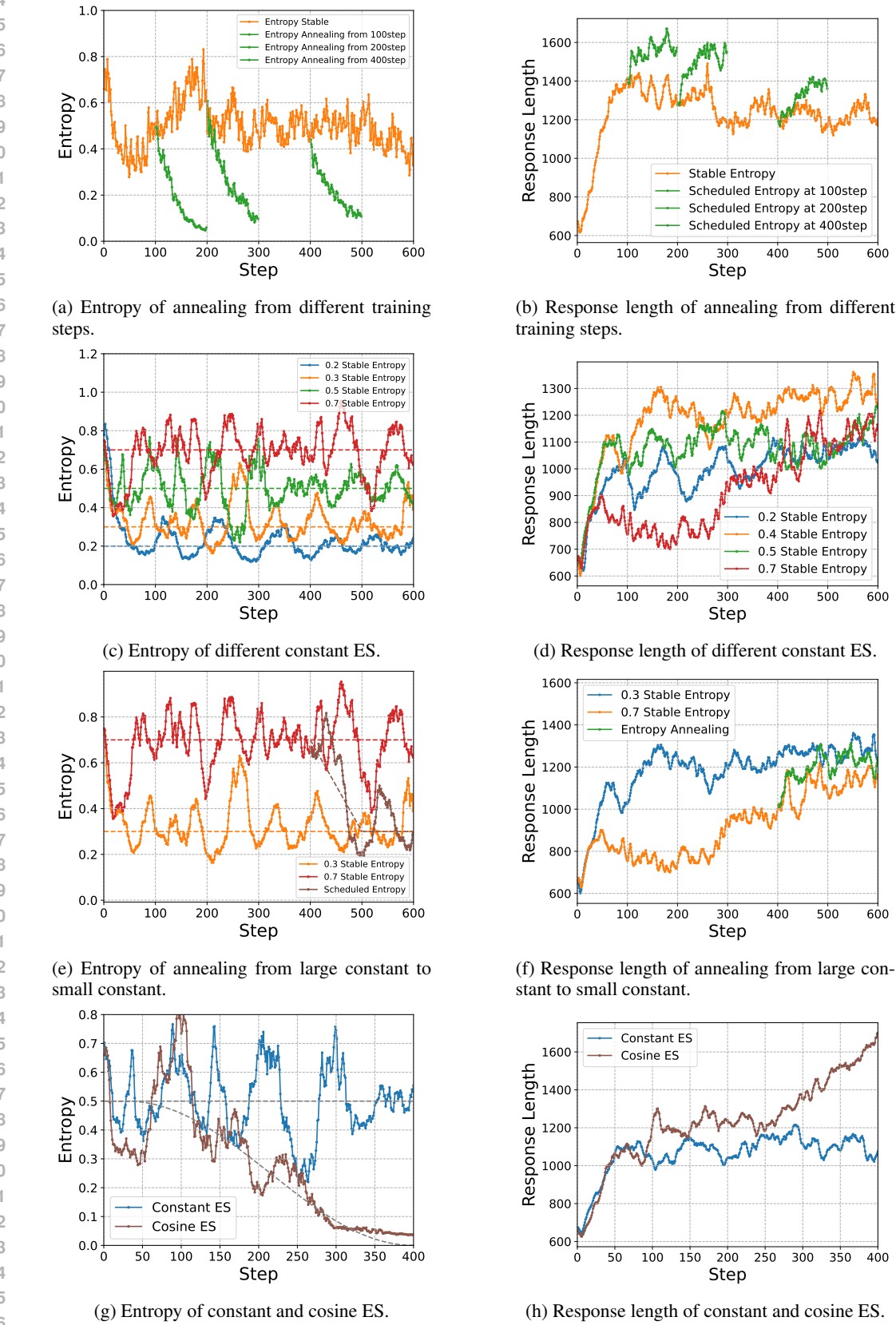

(a) Entropy of annealing from different training steps.

(b) Response length of annealing from different training steps.

(c) Entropy of different constant ES.

(d) Response length of different constant ES.

(e) Entropy of annealing from large constant to small constant.

(f) Response length of annealing from large constant to small constant.

(g) Entropy of constant and cosine ES.

(h) Response length of constant and cosine ES.

Figure 8: The dynamics of response length in the entropy annealing and different ES.

918
919
920
921
922
923
924
925
926
927
928
929
930
931
932
933
934
935
936
937
938
939
940
941
942
943
944
945
946
947
948
949
950
951
952
953
954
955
956
957
958
959
960
961
962
963
964
965
966
967
968
969
970
971

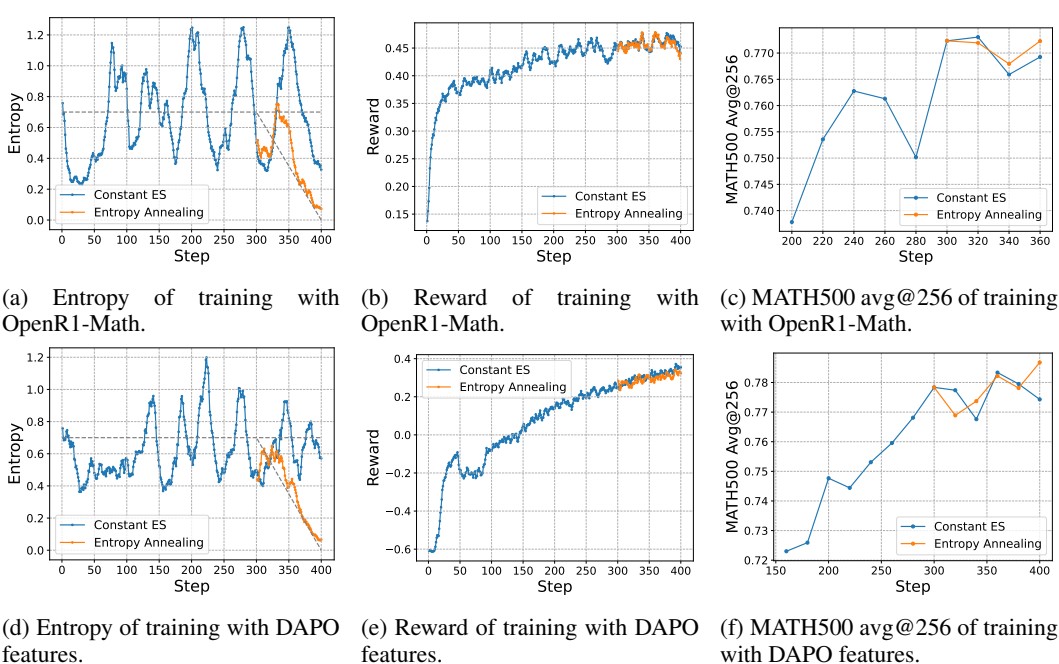

(a) Entropy of training with OpenR1-Math.

(b) Reward of training with OpenR1-Math.

(c) MATH500 avg@256 of training with OpenR1-Math.

(d) Entropy of training with DAPO features.

(e) Reward of training with DAPO features.

(f) MATH500 avg@256 of training with DAPO features.

Figure 9: The entropy, reward and math500 avg@256 of entropy scheduling training with OpenR1-Math dataset and other DAPO RL training features except for clip-higher.

