# OpenReview forum: "Entropy Scheduling in Reinforcement Learning for Large Language Models"
_ICLR.cc/2026/Conference — ICLR 2026 Conference Withdrawn Submission_

### Official Review · Reviewer_B9QM · 2025-10-19

**Soundness:** 2
**Presentation:** 1
**Contribution:** 2
**Rating:** 2
**Confidence:** 3

**Summary:**

The paper explores the role of policy entropy dynamics in reinforcement learning (RL) applied to large language models (LLMs). It proposes Entropy Scheduling (ES) maintaining a high, stable policy entropy early in training to encourage exploration, followed by entropy annealing later to promote convergence. The idea is implemented indirectly by using a PID-based controller. Empirical results on mathematical reasoning benchmarks (AIME, MATH500) suggest some improvements.

**Strengths:**

(1) The paper provides guidelines on how to incorporate the entropy in RL training of LLMs.

(2) Across tasks and model sizes, the reported trend (late-stage entropy annealing helps) appears consistent.

(3) These insights can be integrated efficiently in modern architectures.

**Weaknesses:**

(1) Lack of novelty and depth: The core idea of adjusting entropy over training is not fundamentally new. Entropy regularization, temperature control, and adaptive exploration schedules are long-established in RL (e.g., in Soft Actor-Critic, PPO with temperature annealing). The paper mainly repackages these ideas for LLMs training without introducing a theoretical framework or new algorithmic insight.

(2) Lack of theoretical justification: The proposed “Parallelogram Law of Entropy” is empirically observed without analytical grounding. The paper offers no principled explanation or derivation of why a stable/anneal schedule should be universally optimal. Without a theoretical model, the contribution reads as a collection of heuristics.

(3) Limited experimental scope: All results focus on math and reasoning datasets (AIME, MATH500). There are no experiments on diverse settings such as dialogue or code generation. This undermines claims of generality. Moreover, absolute improvements (e.g., +4% accuracy) are relatively small and may fall within noise margins for large models.

(4) The experiments lack strong baselines (e.g., temperature annealing or PPO entropy bonus decay) and statistical analysis (no variance/error bars). It’s unclear whether reported gains are consistent across seeds or due to tuning advantages.

(5) The paper is not properly formatted into an approach section followed by a results one

**Questions:**

(1) Can the authors relate entropy scheduling to established frameworks such as policy gradient theory or entropy-regularized RL to formalize the intuition?

(2) Have the authors tested the approach on non-reasoning tasks (e.g., dialogue, summarization, code generation) to verify generality?

(3) How does Entropy Scheduling compare against simpler or established baselines such as temperature annealing, entropy-bonus decay, or adaptive KL tuning in PPO?

(4) Can you compute variance or confidence intervals across runs?

(5) The manuscript does not clearly separate the approach and results sections. Could the authors reorganize the paper to delineate the proposed method, implementation details, and experimental results more explicitly?

---

> ### Author Response · Authors · 2025-11-27
> **Response to reviewer B9QM**
>
> Thank you very much for your constructive feedback.
>
> __Response to W1 and Q1__:
> Our core innovation lies in integrating insights from LLM training paradigms, including drawing an analogy to learning rate annealing and incorporating the concept of leveraging additional high-quality data during the annealing phase. Our work is a simple yet effective improvement based on DAPO. It does not undermine the theoretical foundations in reinforcement learning, while this training method can effectively enhance the model's performance.
>
> __Response to W2__:
> Regarding the “Parallelogram Law of Entropy”, we elaborate on its underlying rationale in Section 3.3: “entropy annealing sacrifices performance on larger k (exploration-oriented) to improve performance on smaller k (exploitation-oriented)”.
> Hence, we think the balance between stable and annealing entropy for a whole RL training is theoretically intuitive.
>
> __Response to W3 and Q2__:
> We think the balance of exploration and exploitation is obvious in the RLVR tasks, so we mainly focus on the math tasks. We will extend expand our test domain in the future.
> Notably, the DAPO-32B model achieves a notable 4% performance improvement within only 40 steps of the entropy annealing stage—whereas hundreds of steps are required to attain comparable gains in the stable entropy stage.
>
> __Response to W4, Q3 and Q4__:
> We mainly use GRPO and DAPO as the baselines. We will supplement the other baseline results in future. For statistical, we computed the low-variance and unbiased pass@k metric for all benchmarks, as detailed in Appendix A. All scores were averaged over 128 or 256 model outputs to ensure statistical reliability.
>
> __Response to W5 and Q5__:
> We apologize for any negative impact on the reading experience caused by structural issues. We will refine the paper’s structure in future.

---

### Official Review · Reviewer_4K4W · 2025-10-29

**Soundness:** 3
**Presentation:** 2
**Contribution:** 2
**Rating:** 4
**Confidence:** 2

**Summary:**

The paper studies the impact of different entropy schedules on the performance of LLMs on math problems. It proposes a simple algorithm to enforce an entropy equality constraint by online modification of a hyper-parameter used in the DAPO model (Yu et al., 2025). It then considers several entropy schedules that provide an entropy target to be enforced on the model. These entropy schedules include for instance a constant entropy target or a schedule where the entropy target is increased then decreased in a cyclic manner or decreased only towards the end of training. For each of these schedules, the authors assess the performance of the models mostly by computing the pass@k metric for different values of k. The main conclusion of the paper is that if interested in low k values then a lower entropy target should be enforced and vice versa. It seems also desirable in general to enforce lower entropy towards the end of training.

**Strengths:**

While entropy regularization is well studied in RL, I am not aware of other papers that study such a large diversity of entropy schedules. Entropy is directly related to how explorative an agent is and touches on the central exploration/exploitation dilemma of RL. The topic of entropy scheduling can thus be of great importance to the RL community, even though it remains a primarily heuristic approach to addressing the exploration/exploitation dilemma. Regarding the significance of the results, as the paper is predominantly empirical and as I do not follow the LLM literature, I cannot judge whether the chosen datasets are of sufficient quality and diversity and whether the improvements brought by tuning the entropy schedule are worth the extra parameter search.

**Weaknesses:**

- The take away messages are not that strong or clear. The fact that reducing entropy at the end of training improves performance seems to have already been discussed in other papers as cited in the introduction (l52). The fact that imposing a higher entropy improves diversity of generated solutions and increases the Pass@k metric for higher 'k' is nice though not entirely surprising. Finally, regarding more complex entropy schedulings inspired by learning rate schedules, it wasn't clear from the paper whether they have a clear impact.

- All experiments are performed on three datasets with a single seed with no uncertainty quantification of the stochasticity of the training process. I do not know if this is common in LLM literature but since the paper is entirely focused on the adaptive tuning of a hyper-parameter (the clip-higher $\epsilon$), and since the improvements can sometimes be subtle it raises some questions on the soundness of the drawn conclusions.

- The paper would gain in being a bit more self-contained. The control of the entropy is done not through the entropy weight in Eq. (6) but through the clip-higher method's hyper-parameter (Yu et al., 2025). It is not clear at all for someone unfamiliar with this method how playing with the upper epsilon clip value of PPO can pushe towards higher/lower entropy policies, and it is even less clear why this would be better than controlling the entropy weight.

**Questions:**

- What are the benefits of more complex scheduling schemes like the cosine or cyclic one?
- Why is controlling the entropy better through the clip-higher hyper-parameter than the entropy bonus? Are there benefits in using an entropy schedule if one is not using the clip-higher trick?
- What is the statistical significance of the results?

---

> ### Author Response · Authors · 2025-11-27
> **Response to reviewer 4K4W**
>
> Thank you very much for your constructive feedback.
>
> __Response to W1 and Q1__:
>
> Our core insight is that stable entropy and annealing entropy must be integrated throughout the entire RL training process. In practice, we can directly adopt a cosine entropy scheduling strategy, which offers a balanced solution. Alternatively, we can first maintain entropy within a constant range (0.3–0.5) followed by a 10% entropy annealing phase, as depicted in figure5 and figure6. This dual-stage method has demonstrated effectiveness in enhancing reward signals and final benchmark performance.
>
> __Response to W2 and Q3__:
> For LLMs training, it is uncommon to repeat the full training process using multiple seeds. However, we computed the low-variance and unbiased pass@k metric for all benchmarks, as detailed in Appendix A. All scores were averaged over 128 or 256 model outputs to ensure statistical reliability.
>
> __Response to W3 and Q2__:
> First, we adopt the KL divergence regularization, entropy bonus, and clip-higher methods for entropy control, as presented in Figure 2. Specifically, the KL divergence and entropy bonus methods— which directly introduce weight terms into the objective function—lead to more significant oscillations in the entropy curve and exhibit higher sensitivity to the choice of weighting coefficients. In contrast, the clip-higher method enables smoother regulation of the entropy trajectory. This conclusion is consistent with findings reported in existing literature [1], further validating the effectiveness of the clip-higher method for stable entropy control.
>
> [1] Ganqu Cui, et al. The entropy mechanism of reinforcement learning for. reasoning language models, 2025.

---

### Official Review · Reviewer_7MFQ · 2025-11-01

**Soundness:** 3
**Presentation:** 3
**Contribution:** 3
**Rating:** 4
**Confidence:** 4

**Summary:**

The paper studies the role of policy entropy scheduling in reinforcement learning (RL) and its impact on model performance, particularly in reasoning-oriented tasks. The authors argue that entropy, like the learning rate, plays a crucial role in balancing exploration and exploitation, and propose treating entropy as a schedulable quantity. They show that maintaining stable entropy early in training, followed by entropy annealing, yields better performance. The paper also explores an optimal entropy setting under various objectives, supported by experiments using a PID-based delay control algorithm for dynamic scheduling. The study connects entropy scheduling with metrics such as Pass@k and reward, suggesting principled ways to tune entropy for improved reasoning diversity and stability.

**Strengths:**

- Provides a clear and well-motivated discussion on the role of entropy in reinforcement learning.
- Proposes a practical entropy scheduling approach inspired by learning rate annealing.
- Uses a PID-based control mechanism, grounded in established optimization principles.
- Demonstrates consistent qualitative trends showing improved balance between exploration and exploitation.
- Presents the study in a coherent and well-organized manner, making the motivation and findings accessible.

**Weaknesses:**

- Lack of reported variance or multi-seed runs makes it difficult to assess statistical significance.
- The observed performance improvements are small and sometimes appear within noise levels.
- Some figures (e.g., Figures 3–6) are not clearly interpreted, and result saturation behavior is not well explained.
- The discussion on entropy as a hyperparameter versus an emergent quantity remains somewhat inconsistent.
- Limited analysis on how the optimal entropy schedule generalizes across different problems and settings.

**Questions:**

**Detailed Review:**

The paper explores entropy scheduling as an effective tool for balancing exploration and exploitation in RL. The key idea is that maintaining high entropy at the beginning of training encourages diverse exploration, while gradually annealing it allows the model to consolidate on effective policies later. The analogy between entropy and learning rate scheduling is conceptually appealing, and the empirical evidence generally supports the intuition.

The experiments show that stable early entropy followed by annealing leads to improved performance in several metrics, including Pass@k. The PID-based control algorithm offers a principled way to adjust entropy dynamically, which is a valuable contribution. However, certain claims could be supported more convincingly with quantitative rigor. For example, variance is not reported in Figures 2 through 6, leaving uncertainty about the statistical reliability of the results. It is not clear whether the experiments were repeated across multiple seeds or if the reported numbers correspond to single runs.

In several plots, such as Figures 3 and 4, the differences among methods appear small, and it would be useful to quantify their significance. The paper could also better explain when entropy begins to anneal and whether this threshold (e.g., around 200–300 epochs) is a tunable hyperparameter or task-specific characteristic. Similarly, the discussion around Figure 5 raises questions about how the “optimal constant entropy” is determined and why such an optimum should exist theoretically.

A point of conceptual tension arises in the paper’s discussion of entropy as analogous to the learning rate. Early sections emphasize the similarity, whereas later sections highlight fundamental differences, suggesting that entropy cannot be directly controlled in the same way. Clarifying this distinction and its implications would help strengthen the theoretical argument.

Finally, while the results are promising, they show relatively small numerical improvements. Given the lack of variance and significance testing, it is difficult to assess whether these improvements reflect genuine performance gains or stochastic fluctuations. A more detailed statistical analysis and discussion of stochasticity handling (e.g., across random seeds) would improve the reliability of conclusions.

**Questions:**

1. Are the reported results averaged over multiple seeds, and could variance or confidence intervals be included to assess statistical significance?
2. How is the entropy annealing threshold (e.g., after 200–300 epochs) determined, and is it task-dependent?
3. Can the authors clarify how the PID-based control dynamically adjusts entropy in response to training feedback?
4. In Figure 5, what theoretical justification supports the existence of an “optimal constant entropy” for different Pass@k objectives?
5. How should the results in Figures 3 and 6 be interpreted when differences appear visually small—are they statistically significant?
6. To what extent does entropy scheduling generalize across different environments or RL tasks?
7. Could the authors elaborate on the relationship between entropy and the loss function, and whether entropy can effectively be treated as a tunable hyperparameter?

**Details Of Ethics Concerns:**

None.

---

> ### Author Response · Authors · 2025-11-27
> **Response to reviewer 7MFQ**
>
> Thank you very much for your constructive feedback.
>
> __Response to W1 and Q1__:
>
> We computed the low-variance and unbiased pass@k metric for all benchmarks, as detailed in Appendix A. All scores were averaged over 128 or 256 model outputs to ensure statistical reliability.
>
> __Response to W2 and Q5__:
>
> Reward and performance curves in RL training are typically non-smooth, which is consistent with common observations in RL paradigms. For DAPO-32B, the integration of entropy annealing yields a notable 4% performance improvement, which requires hundreds of training steps in the stable entropy stage to accumulate.
>
> __Response to W3__:
>
> We apologize for not explaining these figures clearly. We explain these figures in detail in each section. In RL training for LLMs, the ascending rate of the reward curve gradually decelerates and eventually approaches a saturation state.
>
> __Response to W4 and Q7__:
>
> We observe that entropy exhibits a similar variation pattern to reward, analogous to how learning rate correlates with validation loss. However, unlike learning rate—a hyperparameter that can be directly preset to a desired trajectory—entropy cannot be configured explicitly. Thus, we regulate the hyperparameters (e.g., clip value) that govern entropy dynamics to maintain entropy in the target state.
>
> __Response to W5 and Q2__:
>
> Our core insight is that stable entropy and annealing entropy must be integrated throughout the entire RL training process. In practice, we can directly adopt a cosine entropy scheduling strategy, which offers a balanced solution. Alternatively, we can first maintain entropy within a constant range (0.3–0.5) followed by a 10% entropy annealing phase, as depicted in figure5 and figure6. This dual-stage method has demonstrated effectiveness in enhancing reward signals and final benchmark performance.
> Although our experiments are primarily conducted on mathematical tasks, we believe this entropy scheduling framework is generalizable across diverse training tasks. For example, many widely used learning rate schedulers are universally applicable in most pre-training scenarios.
>
> __Response to Q3__:
>
> Our adaptive method is described in Lines 317–323. In summary, we dynamically adjust the clip value (increasing or decreasing it) based on feedback from the gap between the preset entropy target and the actual entropy observed during training. This approach is a simple yet effective proportional-integral-derivative (PID) control method.
>
> __Response to Q4__:
> Intuitively, higher entropy allows the model to sustain a higher level of exploration. We contend that increased exploration inherently aligns with a higher k optimization target for the pass@k metric. Our method can effectively control entropy at various constant target values to accommodate different exploration requirements.
>
> __Response to Q6__:
> We think the balance of exploration and exploitation is obvious in the RLVR tasks, so we mainly focus on the math tasks. We will extend expand our test domain in the future.

---

### Official Review · Reviewer_dB91 · 2025-11-02

**Soundness:** 2
**Presentation:** 2
**Contribution:** 2
**Rating:** 2
**Confidence:** 3

**Summary:**

This paper proposes Entropy Scheduling (ES), a training technique for reinforcement learning on large language models (LLMs) that controls the policy entropy over the course of training in an analogous way to learning-rate scheduling. In standard practice, maintaining a high policy entropy (more random exploration) stabilizes training, while letting entropy collapse (rapidly decrease) can lead to quick short-term gains at the cost of exploration. The authors argue these two modes need not be antithetical and can be combined in one run by scheduling entropy. Specifically, they find that keeping entropy high and stable in early training, then annealing (reducing) entropy in a final phase yields better final results than either always-high or always-low entropy. To implement ES, the paper introduces an adaptive clipping mechanism using a simple PID controller to indirectly steer the entropy toward a desired trajectory each step by adjusting the PPO clip parameter. Using this approach, they experiment with different entropy schedules (constant entropy, cosine decay, stable-then-anneal, cyclic, etc.) and identify that a late entropy-annealing schedule consistently leads to higher rewards and accuracy on reasoning benchmarks.

**Strengths:**

1. The paper identifies a key trade-off in RL fine-tuning of LLMs: high entropy fosters exploration but slow convergence, whereas low entropy (entropy collapse) yields quick gains but risks premature convergence.
2. The authors conduct a pilot study comparing three scenarios – entropy annealing from the start, entropy held stable, and entropy annealed at the end – and demonstrate that delayed entropy annealing yields the best final performance on multiple benchmarks.
3. The authors cleverly integrate curriculum learning into the entropy schedule.

**Weaknesses:**

1. A significant weakness is that the paper’s discussion of prior work is limited, missing many relevant references and context. The authors focus mainly on very recent concurrent works (2025 papers on RL for reasoning) but do not adequately situate their approach in the broader reinforcement learning literature. For example, entropy regularization and scheduling have long been studied in RL – Soft Actor-Critic (Haarnoja et al., 2018) famously uses an automatic entropy coefficient adjustment to achieve a target entropy, which is conceptually similar to controlling entropy to balance exploration and exploitation.
2. Building on the above, the core idea of entropy scheduling, while presented as new, feels incremental. The paper’s framing (“entropy is like learning rate”) is novel in phrasing, but in substance it combines existing techniques: using a high entropy bonus or clip range (as done in DAPO) followed by turning those off to let entropy fall – essentially a staged hyperparameter schedule.
3. While the experiments are extensive within the domain of math reasoning benchmarks, the paper’s claims of generality would be stronger if validated on a broader range of tasks.

**Questions:**

1. Results indicate there are optimal entropy values and annealing ratios for different goals (e.g., higher entropy for better Pass@100 vs lower entropy for Pass@1). How would a practitioner choose the right entropy schedule in practice? Is this something that needs manual tuning for each new model or objective, or could an automated heuristic be used (perhaps based on monitoring validation Pass@$k$ during training)?
2. You observed the parallelogram-shaped performance curves under one set of conditions (with Qwen models on math). Have you analyzed why delaying entropy collapse works so well? Is it simply that exploration finds more diverse solutions which then need exploitation to refine, or could it relate to e.g. avoiding reward hacking early on? Additionally, is there any scenario you found where early entropy reduction does outperform (perhaps if the reward is dense and does not require much exploration)?
3. You introduced a PID-based adaptive controller to maintain the entropy schedule. Did you compare this approach to a simpler baseline (for example, manually decreasing the entropy bonus coefficient on a preset schedule without feedback)? It would be useful to know if the adaptive element yields significantly more stable entropy tracking or final performance gains versus open-loop scheduling.

---

> ### Author Response · Authors · 2025-11-27
> **Response to reviewer dB91**
>
> Thank you very much for your constructive feedback.
>
> __Response to W1__:
>
> We apologize for overlooking this relevant work. We also adjust the entropy bonus coefficient to control entropy and compared it with the clip-higher method, as illustrated in figure2. For LLMs, the entropy bonus coefficient exhibits higher sensitivity [1], and the clip-higher method proves more appropriate for RL training of state-of-the-art LLMs. Our research primarily focuses on the integration of stable entropy and annealing entropy in RL training, and our proposed adaptive clip-higher method is simple and could effectively achieve this objective.
>
> __Response to W2__:
>
> We find the entropy has a similar effect with learning rate, and this “hyperparameter schedule” (clip-higehr) is a simple and effective method to control the stable entropy and annealing entropy. Meanwhile, we think the learning rate schedule is also a kind of hyperparameter schedule.
>
> __Response to W3__:
> We think the balance of exploration and exploitation is obvious in the RLVR tasks, so we mainly focus on the math tasks. We will extend expand our test domain in the future.
>
> __Response to Q1__:
> Our core insight is that stable entropy and annealing entropy must be integrated throughout the entire RL training process. In practice, we can directly adopt a cosine entropy scheduling strategy, which offers a balanced solution. Alternatively, we can first maintain entropy within a constant range (0.3–0.5) followed by a 10% entropy annealing phase, as depicted in figure5 and figure6. This dual-stage method has demonstrated effectiveness in enhancing reward signals and final benchmark performance.
>
> __Response to Q2__:
> In RL training for LLMs, the action space is substantially large—this explains why DAPO outperforms GRPO by sustaining high exploration levels. Consequently, maintaining high stable entropy is indispensable for LLMs to generate diverse outputs during RL training. Our work aims to emphasize the significance of the final entropy annealing phase. In section 3.3, we compare the pass@k metrics of models trained with stable entropy alone versus stable entropy combined with annealing entropy. This comparison is justified by the fact that entropy annealing facilitates model convergence, enabling LLMs to generate correct answers with higher probability for solvable questions.
>
> __Response to Q3__:
> We argue that effective entropy control methods should be open-loop; notably, manual reduction of the entropy bonus coefficient still relies on entropy feedback. We conducted comparative experiments on clip-based methods, KL divergence regularization, and entropy bonus strategies, and our findings indicate that clip-based methods yield more stable performance. We will expand this baseline in the future.
>
> [1] The entropy mechanism of reinforcement learning for. reasoning language models, 2025.

---

### Note · Authors · 2026-01-05

I have read and agree with the venue's withdrawal policy on behalf of myself and my co-authors.